# Amortized Decision-Aware
# Bayesian Experimental Design

**Daolang Huang**
Aalto University
daolang.huang@aalto.fi

**Yujia Guo**
Aalto University
yujia.guo@aalto.fi

**Luigi Acerbi**
University of Helsinki
luigi.acerbi@helsinki.fi

**Samuel Kaski**
Aalto University
University of Manchester
samuel.kaski@aalto.fi

## Abstract

Many critical decisions are made based on insights gained from designing, observing, and analyzing a series of experiments. This highlights the crucial role of experimental design, which goes beyond merely collecting information on system parameters as in traditional *Bayesian experimental design* (BED), but also plays a key part in facilitating *downstream decision-making*. Most recent BED methods use an amortized policy network to rapidly design experiments. However, the information gathered through these methods is suboptimal for down-the-line decision-making, as the experiments are not inherently designed with downstream objectives in mind. In this paper, we present an amortized decision-aware BED framework that prioritizes maximizing downstream decision utility. We introduce a novel architecture, the Transformer Neural Decision Process (TNDP), capable of instantly proposing the next experimental design, whilst inferring the downstream decision, thus effectively amortizing both tasks within a unified workflow. We demonstrate the performance of our method across two tasks, showing that it can deliver informative designs and facilitate accurate decision-making[1].

## 1 Introduction

A fundamental challenge in a wide array of disciplines is the design of experiments to infer unknown properties of the systems under study [9, 7]. *Bayesian Experimental Design* (BED) [17, 8, 24, 22] is a powerful framework to guide and optimize experiments by maximizing the expected amount of information about parameters gained from experiments, see Fig. 1(a). To pick the next optimal design, standard BED methods require estimating and optimizing the expected information gain (EIG) over the design space, which can be extremely time-consuming. This limitation has led to the development of *amortized* BED [10, 14, 5, 6], a *policy-based* method which leverages a neural network policy trained on simulated experimental trajectories to quickly generate designs, as illustrated in Fig. 1(b).

However, the ultimate goal in many tasks extends beyond parameter inference to inform a *downstream decision-making* task by leveraging our understanding of these parameters, such as in personalized medical diagnostics [4]. Previous amortized BED methods do not take down-the-line decision-making tasks into account, which is suboptimal for decision-making in scenarios where experiments can be adaptively designed. *Loss-calibrated inference*, which was originally introduced by Lacoste-Julien

---

[1]The full version of this work can be found at: https://arxiv.org/abs/2411.02064.

Workshop on Bayesian Decision-making and Uncertainty, 38th Conference on Neural Information Processing Systems (NeurIPS 2024).

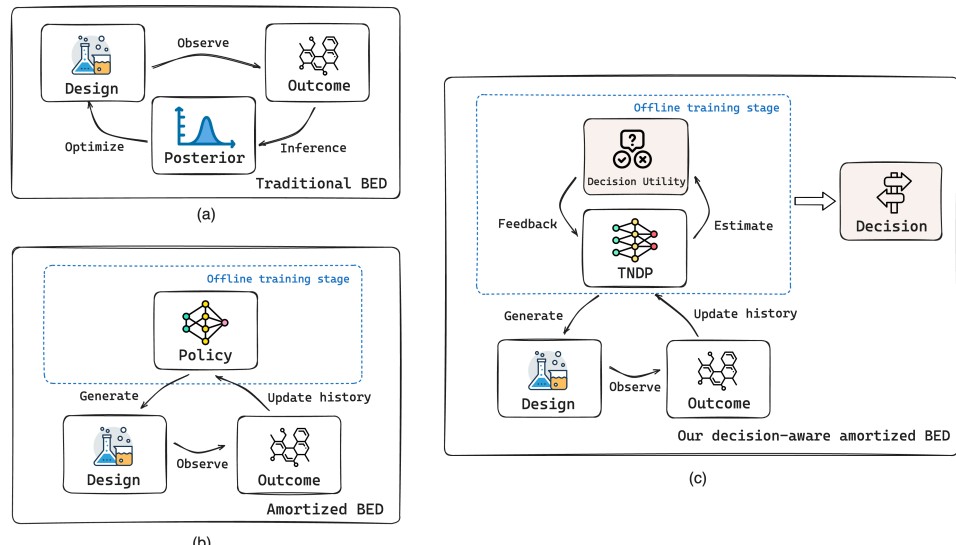

Figure 1: **Overview of BED workflows.** (a) Traditional BED, which iterates between optimizing designs, running experiments, and updating the model via Bayesian inference. (b) Amortized BED, which uses a policy network for rapid experimental design generation. (c) Our decision-aware amortized BED integrates decision utility in the training to facilitate downstream decision-making.

et al. [16] for variational approximations in Bayesian inference, provides a concept that adjusts the inference process to capture posterior regions critical for decision-making tasks. Inspired by this concept, we consider integrating decision-making directly into the experimental design process to align the proposed experimental designs more closely with the ultimate decision-making task.

In this paper, we propose an amortized decision-making-aware BED framework, see Fig. 1(c). We identify two key aspects where previous amortized BED methods fall short when applied to downstream decision-making tasks. First, the training objective of the existing methods does not consider downstream decision tasks. Therefore, we introduce the concept of *Decision Utility Gain* (DUG) to guide experimental design to better align with the downstream objective. Second, to obtain the optimal decision, we still need to explicitly approximate the predictive distribution of the outcomes to estimate the utility. Current amortized methods learn this distribution only implicitly and therefore do not fully amortize the decision-making process. To address this, we propose a novel *Transformer neural decision process* (TNDP) architecture, where the system can instantly propose informative experimental designs and make final decisions. Finally, we train under a non-myopic objective function that ensures decisions are made with consideration of future outcomes. We empirically show the effectiveness of our method through two tasks.

## 2 Decision-aware BED

### 2.1 Preliminaries and problem setup

In this paper, we consider scenarios in which we design a series of experiments $\xi \in \Xi$ and observe corresponding outcomes $y$ to inform a final decision-making step. The experimental history is denoted as $h_{1:t} = \{(\xi_1, y_1), ..., (\xi_t, y_t)\}$ and we assume a fixed experimental budget with $T$ query steps. Our objective is to identify an optimal decision $a^*$ from a set of possible decisions $\mathcal{A}$ at time $T$.

To make decisions under uncertainty, *Bayesian decision theory* [3] provides an axiomatic framework by incorporating probabilistic beliefs about unknown parameters into decision-making. Given a task-specific utility function $u(\theta, a)$, which quantifies the value of the outcomes from different decisions $a \in \mathcal{A}$ when the system is in state $\theta$, the optimal decision is then determined by maximizing the expected utility under the posterior distribution of the parameters $p(\theta|h_{1:t})$.

In many real-world scenarios, outcomes are stochastic and it is more typical to make decisions based on their *predictive distribution* $p(y|\xi, h_{1:t}) = \mathbb{E}_{p(\theta|h_{1:t})}[p(y|\xi, \theta, h_{1:t})]$, such as in clinical

trials where the optimal treatment is chosen based on predicted patient responses. A similar setup can be found in [15, 28]. Thus, we can represent our belief directly as $p(y_\Xi|h_{1:t}) \equiv \{p(y|\xi, h_{1:t})\}_{\xi \in \Xi}$, which is a stochastic process that defines a joint predictive distribution of outcomes indexed by the elements of the design set $\Xi$, given the current information $h_{1:t}$. Our utility function is then expressed as $u(y_\Xi, a)$, which is a natural extension of the traditional definition of utility by marginalizing out the posterior distribution of $\theta$. The rule for making the optimal decision is then reformulated as:

$$a^* = \arg\max_{a \in A} \mathbb{E}_{p(y_\Xi|h_{1:t})}[u(y_\Xi, a)]. \tag{1}$$

## 2.2 Decision utility gain

To quantify the effectiveness of each experimental design in terms of decision-making, we introduce *Decision Utility Gain* (DUG), which is defined as the difference in the expected utility of the best decision, with the new information obtained from the current experimental design, versus the best decision with the information obtained from previous experiments.

**Definition 2.1.** Given a historical experimental trajectory $h_{1:t-1}$, the *Decision Utility Gain* (DUG) for a given design $\xi_t$ and its corresponding outcome $y_t$ at step $t$ is defined as follows:

$$\mathrm{DUG}(\xi_t, y_t) = \max_{a \in A} \mathbb{E}_{p(y_\Xi|h_{1:t-1} \cup \{(\xi_t, y_t)\})}[u(y_\Xi, a)] - \max_{a \in A} \mathbb{E}_{p(y_\Xi|h_{1:t-1})}[u(y_\Xi, a)]. \tag{2}$$

DUG measures the improvement in the *maximum* expected utility from observing a new experimental design, differing in this from standard marginal utility gain (see e.g., [12]). The optimal design is the one that provides the largest increase in maximal expected utility. At the time we choose the design $\xi_t$, the outcome remains uncertain. Therefore, we should consider the *Expected Decision Utility Gain* (EDUG) to select the next design, which is defined as $\mathrm{EDUG}(\xi_t) = \mathbb{E}_{p(y_t|\xi_t, h_{1:t-1})}[\mathrm{DUG}(\xi_t, y_t)]$. The one-step lookahead optimal design can be determined by maximizing EDUG with $\xi^* = \arg\max_{\xi \in \Xi} \mathrm{EDUG}(\xi)$. However, in practice, the true predictive distributions are often unknown, making the optimization of EDUG exceptionally challenging. This difficulty arises due to the inherent bi-level optimization problem and the need to evaluate two layers of expectations.

To avoid the expensive cost of optimizing EDUG, we propose using a policy network that directly maps historical data to the next design. This approach sidesteps the need to iteratively optimize EDUG by learning a design strategy over many simulated experiment trajectories beforehand.

## 2.3 Amortization with TNDP

Our architecture, termed *Transformer Neural Decision Process* (TNDP), is a novel architecture building upon the Transformer neural process (TNP) [20]. It aims to amortize both the experimental design and the subsequent decision-making. A general introduction to TNP can be found in Appendix A.

The data architecture of our system comprises four parts: A **context set** $D^{(c)} = \{(\xi_i^{(c)}, y_i^{(c)})\}_{i=1}^{t}$ contains all past $t$-step designs and outcomes; A **prediction set** $D^{(p)} = \{(\xi_i^{(p)}, y_i^{(p)})\}_{i=1}^{n_p}$ consists of $n_p$ design-outcome pairs used for approximating $p(y_\Xi|h_{1:t})$. The output from this head can then be used to estimate the expected utility; A **query set** $D^{(q)} = \{\xi_i^{(q)}\}_{i=1}^{n_q}$ consists of $n_q$ candidate experimental designs being considered for the next step; **Global information** $\mathrm{GI} = [t, \gamma]$ where $t$ represents the current step, and $\gamma$ encapsulates task-related information, which could include contextual data relevant to the decision-making process.

TNDP comprises four main components, the full architecture is shown in Fig. 2(a). At first, the **data embedder block** $f_{\mathrm{emb}}$ maps each set of $D$ to an aligned embedding space. The embeddings are then concatenated to form a unified representation $\boldsymbol{E} = \mathrm{concat}(\boldsymbol{E}^{(c)}, \boldsymbol{E}^{(p)}, \boldsymbol{E}^{(q)}, \boldsymbol{E}^{\mathrm{GI}})$. After the initial embedding, the **Transformer block** $f_{\mathrm{tfm}}$ processes $\boldsymbol{E}$ using attention mechanisms that allow for selective interactions between different data components, ensuring that each part contributes appropriately to the final output. Fig. 2(b) shows an example attention mask. The output of $f_{\mathrm{tfm}}$ is then split according to the specific needs of the query and prediction head $\boldsymbol{\lambda} = [\boldsymbol{\lambda}^{(p)}, \boldsymbol{\lambda}^{(q)}] = f_{\mathrm{tfm}}(\boldsymbol{E})$.

The primary role of the **prediction head** $f_p$ is to approximate $p(y_\Xi|h_{1:t})$ at any step $t$ with a family of parameterized distributions $q(y_\Xi|\boldsymbol{p}_t)$, where $\boldsymbol{p}_t = f_p(\boldsymbol{\lambda}_t^{(p)})$ is the output of $f_p$. We choose a Gaussian

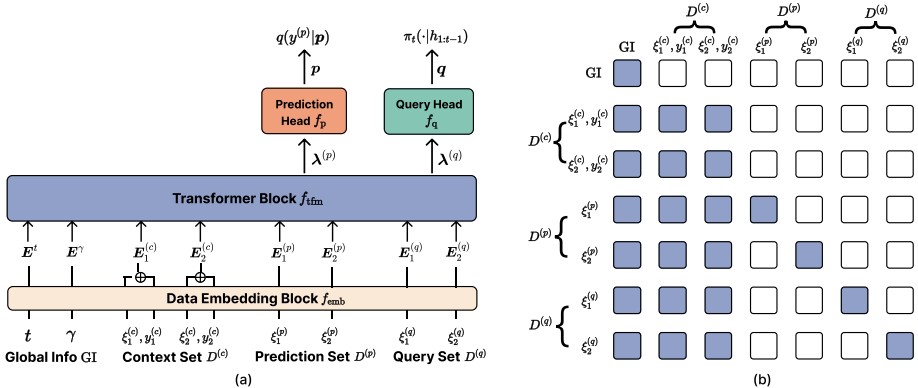

Figure 2: **Illustration of TNDP.** (a) An overview of TNDP architecture with input consisting of 2 observed design-outcome pairs from $D^{(c)}$, 2 designs from $D^{(p)}$ for prediction, and 2 candidate designs from $D^{(q)}$ for query. (b) The corresponding attention mask. The colored squares indicate that the elements on the left can attend to the elements on the top in the self-attention layer of $f_{\text{tfm}}$.

likelihood and train $f_{\text{p}}$ by minimizing the negative log-likelihood of the predicted probabilities:

$$\mathcal{L}^{(\text{p})} = -\sum_{t=1}^{T}\sum_{i=1}^{n_{\text{p}}} \log q(y_i^{(\text{p})}|\boldsymbol{p}_{i,t}) = -\sum_{t=1}^{T}\sum_{i=1}^{n_{\text{p}}} \log \mathcal{N}(y_i^{(\text{p})}|\boldsymbol{\mu}_{i,t}, \boldsymbol{\sigma}_{i,t}^2), \tag{3}$$

where $\boldsymbol{p}_{i,t}$ represents the output of design $\xi_i^{(\text{p})}$ at step $t$, $\boldsymbol{\mu}$ and $\boldsymbol{\sigma}$ are the predicted mean and standard deviation split from $\boldsymbol{p}$.

Lastly, the **query head** $f_{\text{q}}$ processes the embeddings $\boldsymbol{\lambda}^{(\text{q})}$ from the Transformer block to generate a policy distribution over possible experimental designs. The outputs of the query head, $\boldsymbol{q} = f_{\text{q}}(\boldsymbol{\lambda}^{(\text{q})})$, are mapped to a probability distribution $\pi(\xi_t^{(\text{q})}|h_{1:t-1})$ via a Softmax function. To design a reward signal that guides the query head $f_{\text{q}}$ in proposing informative designs, we first define a single-step immediate reward based on DUG (Eq. (2)), replacing the true predictive distribution with our approximated distribution:

$$r_t(\xi_t^{(\text{q})}) = \max_{a \in A} \mathbb{E}_{q(y_\Xi|\boldsymbol{p}_t)} \left[u(y_\Xi, a)\right] - \max_{a \in A} \mathbb{E}_{q(y_\Xi|\boldsymbol{p}_{t-1})} \left[u(y_\Xi, a)\right]. \tag{4}$$

This reward quantifies how the experimental design influences our decision-making by estimating the improvement in expected utility that results from incorporating new experimental outcomes. However, this objective remains myopic, as it does not account for the future or the final decision-making. To address this, we employ the REINFORCE algorithm [30]. The final loss of $f_{\text{q}}$ can be written as:

$$\mathcal{L}^{(\text{q})} = -\sum_{t=1}^{T} R_t \log \pi(\xi_t^{(\text{q})}|h_{1:t-1}), \tag{5}$$

where $R_t = \sum_{k=t}^{T} \alpha^{k-t} r_k(\xi_k^{(\text{q})})$ represents the non-myopic discounted reward. The discount factor $\alpha$ is used to decrease the importance of rewards received at later time step. $\xi_t^{(\text{q})}$ is obtained through sampling from the policy distribution $\xi_t^{(\text{q})} \sim \pi(\cdot|h_{1:t-1})$. The details of implementing and training TNDP are shown in Appendix B.

## 3 Experiments

### 3.1 Toy example: targeted active learning

We begin with an illustrative example to show how our TNDP works. We consider a synthetic regression task where the goal is to perform regression at a specific test point $x^*$ on an unknown function. To accurately predict this point, we need to actively collect some new points to query.

The design space $\Xi = \mathcal{X}$ is the domain of $x$, and $y$ is the corresponding noisy observations of the function. Let $\mathcal{Q}(\mathcal{X})$ denote the set of combinations of distributions that can be output by TNDP, we can then define decision space to be $\mathcal{A} = \mathcal{Q}(\mathcal{X})$. The downstream decision is to output a predictive distribution for $y^*$ given a test point $x^*$, and the utility function $u(y_\Xi, a) = \log q(y^*|x^*, h_{1:t})$ is the log probability of $y$ under the predicted distribution.

During training, we sample functions from Gaussian Processes (GPs) [23] with squared exponential kernels of varying output variances and lengthscales and randomly sample a point as the test point $x^*$. We set the global contextual information $\lambda$ as the test point $x^*$. For illustration purposes, we consider only the case where $T = 1$. Additional details for the data generation can be found in Appendix C.

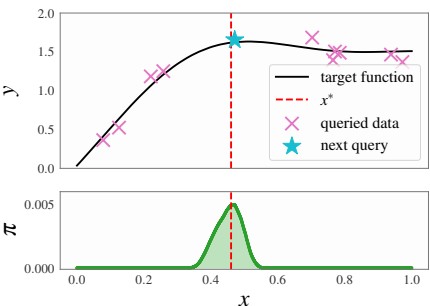

Figure 3: **Results of toy example.** The top figure represents the true function and the initial known points. The red line indicates the location of $x^*$. The blue star marks the next query point, sampled from the policy's predicted distribution shown in the bottom figure.

**Results.** From Fig. 3, we can observe that the values of $\pi$ concentrate near $x^*$, meaning our query head $f_q$ tends to query points close to $x^*$ to maximize the DUG. This is an intuitive example demonstrating that our TNDP can adjust its design strategy based on the downstream task.

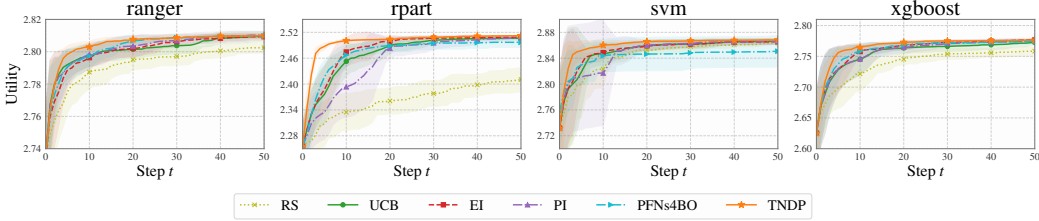

Figure 4: **Average utility on Top-$k$ HPO task.** The error bars represent the standard deviation over five runs. **TNDP consistently achieved the best performance regarding the utility.**

## 3.2 Top-$k$ hyperparameter optimization

In traditional optimization tasks, we typically only aim to find a single point that maximizes the underlying function $f$. However, instead of identifying a single optimal point, there are scenarios where we wish to estimate a set of top-$k$ distinct optima, such as in materials discovery [18, 27].

In this experiment, we choose hyperparameter optimization (HPO) tasks and conduct experiments on the HPO-B datasets [1]. The design space $\Xi \subseteq \mathcal{X}$ is a finite set defined over the hyperparameter space and the outcome $y$ is the accuracy of a given configuration. Our decision is to find $k$ hyperparameter sets, denoted as $a = (a_1, ..., a_k) \in A \subseteq \mathcal{X}^k$, with $a_i \neq a_j$. The utility function is then defined as $u(y_\Xi, a) = \sum_{i=1}^{k} y_{a_i}$, where $y_{a_i}$ is the accuracy corresponding to the configuration $a_i$.

We compare our methods with five different BO acquisition functions: random sampling (RS), Upper Confidence Bound (UCB), Expected Improvement (EI), Probability of Improvement (PI), and an amortized method PFNs4BO [19]. We set $k = 3$ and $T = 50$. Our experiments are conducted on four search spaces. All results are evaluated on a predefined test set. For more details, see Appendix D.

**Results.** From the results (Fig. 4), our method demonstrates superior performance across all four meta-datasets, particularly during the first 10 queries.

## 4 Discussion and conclusion

In this paper, we introduced a decision-aware amortized BED framework with a novel TNDP architecture to optimize experimental designs for better decision-making. Future work includes conducting more extensive empirical tests and ablation studies, deploying more advanced RL algorithms [26] to enhance training stability, addressing robust experimental design under model misspecification [22, 13, 25], and developing dimension-agnostic methods to expand the scope of amortization.

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

# Appendix

## A  Conditional neural processes

CNPs [11] are designed to model complex stochastic processes through a flexible architecture that utilizes a *context set* and a *target set*. The context set consists of observed data points that the model uses to form its understanding, while the target set includes the points to be predicted. The traditional CNP architecture includes an encoder and a decoder. The encoder is a DeepSet architecture to ensure permutation invariance, it transforms each context point individually and then aggregates these transformations into a single representation that captures the overall context. The decoder then uses this representation to generate predictions for the target set, typically employing a Gaussian likelihood for approximation of the true predictive distributions. Due to the analytically tractable likelihood, CNPs can be efficiently trained through maximum likelihood estimation.

### A.1  Transformer neural processes

Transformer Neural Processes (TNPs), introduced by [20], enhance the flexibility and expressiveness of CNPs by incorporating the Transformer's attention mechanism [29]. In TNPs, the transformer architecture uses self-attention to process the context set, dynamically weighting the importance of each point. This allows the model to create a rich representation of the context, which is then used by the decoder to generate predictions for the target set. The attention mechanism in TNPs facilitates the handling of large and variable-sized context sets, improving the model's performance on tasks with complex input-output relationships. The Transformer architecture is also useful in our setups where certain designs may have a more significant impact on the decision-making process than others. For more details about TNPs, please refer to [20].

## B  Additional details of TNDP

### B.1  Full algorithm for training TNDP

---
**Algorithm 1** Transformer Neural Decision Processes (TNDP)
---
1: **Input:** Utility function $u(y_\Xi, a)$, prior $p(\theta)$, likelihood $p(y|\theta, \xi)$, query horizon $T$
2: **Output:** Trained TNDP
3: **while** within the training budget **do**
4:     Sample $\theta \sim p(\theta)$ and initialize $D$
5:     **for** $t = 1$ to $T$ **do**
6:         $\xi_t^{(q)} \sim \pi_t(\cdot|h_{1:t-1})$           ▷ sample next design from policy
7:         Sample $y_t \sim p(y|\theta, \xi)$           ▷ observe outcome
8:         Set $h_t = h_{t-1} \cup \{(\xi_t^{(q)}, y_t)\}$         ▷ update history
9:         Set $D^{(c)} = h_{1:t}, D^{(q)} = D^{(q)} \setminus \{\xi_t^{(q)}\}$     ▷ update $D$
10:        Calculate $r_t(\xi_t^{(q)})$ with $u(y_\Xi, a)$ using Eq. (4)     ▷ calculate reward
11:     **end for**
12:     $R_t = \sum_{k=t}^{T} \alpha^{k-t} r_k(\xi_k^{(q)})$         ▷ calculate cumulative reward
13:     Update TNDP using $\mathcal{L}^{(p)}$ (Eq. (3)) and $\mathcal{L}^{(q)}$ (Eq. (5))
14: **end while**
15: At deployment, we can use $f^{(q)}$ to sequentially query $T$ designs. Afterward, based on the queried experiments, we perform one-step final decision-making using the prediction from $f^{(p)}$.

---

### B.2  Embedders

The embedder $f_{\text{emb}}$ is responsible for mapping the raw data to a space of the same dimension. For the toy example and the top-$k$ hyperparameter task, we use three embedders: a design embedder $f_{\text{emb}}^{(\xi)}$, an outcome embedder $f_{\text{emb}}^{(y)}$, and a time step embedder $f_{\text{emb}}^{(t)}$. Both $f_{\text{emb}}^{(\xi)}$ and $f_{\text{emb}}^{(y)}$ are multi-layer perceptions (MLPs) with the following architecture:

- **Hidden dimension**: the dimension of the hidden layers, set to 32.
- **Output dimension**: the dimension of the output space, set to 32.
- **Depth**: the number of layers in the neural network, set to 4.
- **Activation function**: ReLU is used as the activation function for the hidden layers.

The time step embedder $f_{\text{emb}}^{(t)}$ is a discrete embedding layer that maps time steps to a continuous embedding space of dimension 32.

For the decision-aware active learning task, since the design space contains both the covariates and the decision, we use four embedders: a covariate embedder $f_{\text{emb}}^{(x)}$, a decision embedder $f_{\text{emb}}^{(d)}$, an outcome embedder $f_{\text{emb}}^{(y)}$, and a time step embedder $f_{\text{emb}}^{(t)}$. $f_{\text{emb}}^{(x)}$, $f_{\text{emb}}^{(y)}$ and $f_{\text{emb}}^{(t)}$ are MLPs which use the same settings as described above. The decision embedder $f_{\text{emb}}^{(d)}$ is another discrete embedding layer.

For context embedding $\boldsymbol{E}^{(\text{c})}$, we first map each $\xi_i^{(\text{c})}$ and $y_i^{(\text{c})}$ to the same dimension using their respective embedders, and then sum them to obtain the final embedding. For prediction embedding $\boldsymbol{E}^{(\text{p})}$ and query embedding $\boldsymbol{E}^{(\text{q})}$, we only encode the designs. For $\boldsymbol{E}^{\text{GI}}$, except the embeddings of the time step, we also encode the global contextual information $\lambda$ using $f_{\text{emb}}^{(x)}$ in the toy example and the decision-aware active learning task. All the embeddings are then concatenated together to form our final embedding $\boldsymbol{E}$.

### B.3 Transformer blocks

We utilize the official `TransformerEncoder` layer of PyTorch [21] (https://pytorch.org) for our transformer architecture. For all experiments, we use the same configuration, which is as follows:

- **Number of layers**: 6
- **Number of heads**: 8
- **Dimension of feedforward layer**: 128
- **Dropout rate**: 0.0
- **Dimension of embedding**: 32

### B.4 Output heads

The prediction head, $f_{\text{p}}$ is an MLP that maps the Transformer's output embeddings of the query set to the predicted outcomes. It consists of an input layer with 32 hidden units, a ReLU activation function, and an output layer. The output layer predicts the mean and variance of a Gaussian likelihood, similar to CNPs.

For the query head $f_{\text{q}}$, all candidate experimental designs are first mapped to embeddings $\boldsymbol{\lambda}^{(\text{q})}$ by the Transformer, and these embeddings are then passed through $f_{\text{q}}$ to obtain individual outputs. We then apply a Softmax function to these outputs to ensure a proper probability distribution. $f_{\text{q}}$ is an MLP consisting of an input layer with 32 hidden units, a ReLU activation function, and an output layer.

### B.5 Training details

For all experiments, we use the same configuration to train our model. We set the initial learning rate to `5e-4` and employ the cosine annealing learning rate scheduler. The number of training epochs is set to 50,000. For the REINFORCE algorithm, we select a discount factor of $\alpha = 0.99$.

## C Details of toy example

### C.1 Data generation

In our toy example, we generate data using a GP with the Squared Exponential (SE) kernel, which is defined as:

$$k(x, x') = v \exp\left(-\frac{(x-x')^2}{2\ell^2}\right), \tag{A.1}$$

where $v$ is the variance, and $\ell$ is the lengthscale. Specifically, in each training iteration, we draw a random lengthscale $\ell \sim 0.25 + 0.75 \times U(0,1)$ and the variance $v \sim 0.1 + U(0,1)$, where $U(0,1)$ denotes a uniform random variable between 0 and 1.

# D  Details of top-$k$ hyperparameter optimization experiments

## D.1  Data

In this task, we use HPO-B benchmark datasets [1]. The HPO-B dataset is a large-scale benchmark for HPO tasks, derived from the OpenML repository. It consists of 176 search spaces (algorithms) evaluated on 196 datasets, with a total of 6.4 million hyperparameter evaluations. This dataset is designed to facilitate reproducible and fair comparisons of HPO methods by providing explicit experimental protocols, splits, and evaluation measures.

We extracted four meta-datasets from the HPOB dataset: ranger (7609), svm (5891), rpart (5859), and xgboost (5971). For detailed information on the datasets, please refer to `https://github.com/releaunifreiburg/HPO-B`.

## D.2  Other methods description

In our experiments, we compare our method with several common acquisition functions used in HPO. We use GPs as surrogate models for these acquisition functions. All the implementations are based on BoTorch [2] (`https://botorch.org/`). The acquisition functions compared are as follows:

- **Random Sampling (RS)**: This method selects hyperparameters randomly from the search space, without using any surrogate model or acquisition function.
- **Upper Confidence Bound (UCB)**: This acquisition function balances exploration and exploitation by selecting points that maximize the upper confidence bound. The UCB is defined as:
$$\alpha_{\text{UCB}}(\mathbf{x}) = \mu(\mathbf{x}) + \kappa\sigma(\mathbf{x}), \tag{A.2}$$
where $\mu(\mathbf{x})$ is the predicted mean, $\sigma(\mathbf{x})$ is the predicted standard deviation, and $\kappa$ is a parameter that controls the trade-off between exploration and exploitation.
- **Expected Improvement (EI)**: This acquisition function selects points that are expected to yield the greatest improvement over the current best observation. The EI is defined as:
$$\alpha_{\text{EI}}(\mathbf{x}) = \mathbb{E}[\max(0, f(\mathbf{x}) - f(\mathbf{x}^+))], \tag{A.3}$$
where $f(\mathbf{x}^+)$ is the current best value observed, and the expectation is taken over the predictive distribution of $f(\mathbf{x})$.
- **Probability of Improvement (PI)**: This acquisition function selects points that have the highest probability of improving over the current best observation. The PI is defined as:
$$\alpha_{\text{PI}}(\mathbf{x}) = \Phi\left(\frac{\mu(\mathbf{x}) - f(\mathbf{x}^+) - \xi}{\sigma(\mathbf{x})}\right), \tag{A.4}$$
where $\Phi$ is the cumulative distribution function of the standard normal distribution, $f(\mathbf{x}^+)$ is the current best value observed, and $\xi$ is a parameter that encourages exploration.

We also compared our method with an amortized method **PFNs4BO** [19]. It is a Transformer-based model designed for hyperparameter optimization which does not consider the downstream task. We used the pre-trained PFNs4BO-BNN model and chose PI as the acquisition function. We used the PFNs4BO's official implementation (`https://github.com/automl/PFNs4BO`).

