# OpenReview forum: "Amortized Decision-Aware Bayesian Experimental Design"
_NeurIPS.cc/2024/Workshop/BDU — NeurIPS BDU Workshop 2024 Poster_

### Official Review · Reviewer_vWVS · 2024-09-28

**Rating:** 6
**Confidence:** 3

**Review:**

The paper introduces an innovative approach to Bayesian Experimental Design (BED), termed Amortized Decision-Aware BED. The key novelty lies in the integration of decision-making objectives directly into the experiment design process, ensuring that experimental designs maximize downstream decision utility.

**Strengths**
The method seem to be novel, as most prior BED methods focus purely on parameter inference without explicitly considering the decision-making tasks. TNDP architecture design is sensible by utilizing transformer and special masking designs. Experimental results on two tasks: a toy example of targeted active learning and a more complex Top-k Hyperparameter Optimization task. The TNDP demonstrates superior performance in terms of utility maximization compared to several baseline methods, including Random Sampling, UCB, EI, PI, and PFNs4BO.

**Weakness**
Although the idea is interesting, architectural novelty of TNDP is very limited, as well as the theoritical results on the algorithm (soundness and does it actually maximize EDUG). Emperical results are good for lower steps, but it would be a stronger paper if more emperical results are provided since this is an emperical paper.

---

### Official Review · Reviewer_HmWG · 2024-09-29
**An Interesting Concept of Decision-Aware Bayesian Experimental Design, but Lacks Strong Empirical Justification**

**Rating:** 4
**Confidence:** 4

**Review:**

This paper introduces a novel approach to Bayesian Experimental Design (BED) by directly incorporating downstream decision-making utility into the design process. The approach is well-motivated by Bayesian decision theory, and the authors propose an amortized architecture to predict experimental designs that maximize a non-myopic, discounted cumulative Decision Utility Gain (DUG), which is a nice to the literature. However, while the framing is appealing, the paper has several shortcomings in its empirical validation and comparisons with relevant baselines.

Strengths:

- The paper presents a clean and novel setup that directly considers downstream decision-making utility in the experimental design process, which is an innovative contribution.
- The decision-aware framing is a compelling extension of Bayesian decision theory, and the concept of Decision Utility Gain (DUG) is well justified within this framework.
- The authors successfully motivate the challenge of optimizing DUG and propose a Transformer Neural Decision Process (TNDP) to bypass the complexity of traditional lookahead optimization methods, which is a creative architectural choice.

Weaknesses:

- Marginal improvement: The proposed method shows no statistically significant improvement compared to baseline methods, with only marginal gains observed. The lack of substantial empirical improvement undermines the paper's claim of its practical relevance.
- Inadequate experimental comparisons: The experimentation feels incomplete and lacks an apple-to-apple comparison with similarly motivated lookahead acquisition functions, such as the Knowledge Gradient (KG). Since TNDP aims to bypass the complexity of optimizing EDUG by directly predicting the next design, it should have been compared to lookahead acquisition functions, which also perform a form of lookahead optimization.
- Pretraining and transfer learning: The proposed TNDP architecture requires pretraining, but the paper does not adequately discuss how the pretraining data could be leveraged by GP-based baselines, for example, through transfer learning or task-specific priors.
- Overly simple decision problem: Despite the paper’s emphasis on decision-making, the decision problem it tackles seems overly simple, which raises the question of whether the proposed method offers significant advantages over standard Bayesian Optimization (BO) methods. It is unclear how the decision-aware approach diverges significantly from a regular optimization problem that current BO methods could handle adequately.
- Limited diversity in experiments: While the paper includes four experiments, they are all variations of Hyperparameter Optimization (HPO) problems, offering little variety in experimental tasks. This lack of diversity in experimentation weakens the generalizability of the proposed method.
- No reported training time: The training process for pretrained models like TNDP is arguably non-trivial in terms of computational resources and time, yet the paper does not report any details on the training time. This omission leaves an important factor in the practical applicability of the method unaddressed.

---

### Decision · Program_Chairs · 2024-10-09

Accept (Poster)